# Speculative Thinking: Enhancing Small-Model Reasoning with Large Model Guidance at Inference Time

**Wang Yang[1], Xiang Yue[2], Vipin Chaudhary[1], Xiaotian Han[1]**
[1]Case Western Reserve University  [2]Carnegie Mellon University
{wxy320,vxc204,xhan}@case.edu  xyue2@andrew.cmu.edu

## Abstract

Recent advances leverage post-training to enhance model reasoning performance, which typically requires costly training pipelines and still suffers from inefficient, overly lengthy outputs. We introduce *Speculative Thinking*[1], a training-free framework that enables large reasoning models to guide smaller ones during inference at the reasoning level, distinct from speculative decoding, which operates at the token level. Our approach is based on two observations: (1) reasoning-supportive tokens such as *"wait"* frequently appear after structural delimiters like "\n\n", serving as signals for reflection or continuation; and (2) larger models exhibit stronger control over reflective behavior, reducing unnecessary backtracking while improving reasoning quality. By strategically delegating reflective steps to a more capable model, our method significantly boosts the reasoning accuracy of reasoning models while shortening their output. With the assistance of the 32B reasoning model, the 1.5B model's accuracy on MATH500 increases from 83.2% to 89.4%, marking a substantial improvement of 6.2%. Simultaneously, the average output length is reduced from 5439 tokens to 4583 tokens, representing a 15.7% decrease. Moreover, when applied to a non-reasoning model (Qwen-2.5-7B-Instruct), our framework boosts its accuracy from 74.0% to 81.8% on the same benchmark, achieving a relative improvement of 7.8%.

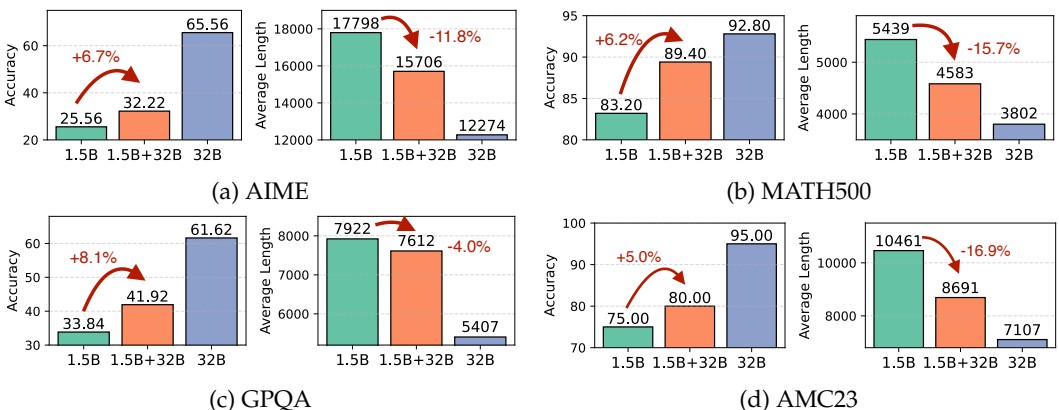

Figure 1: Speculative Thinking significantly improves the 1.5B model's reasoning accuracy while simultaneously reducing its average output length. This figure compares the accuracy and average output length of models on four mathematical and reasoning datasets, including AIME 2020–2024, MATH500, GPQA, and AMC23. "1.5B" denotes the Deepseek-Distilled Qwen 2.5-1.5B model, "32B" refers to the Deepseek-Distilled Qwen 2.5-32B model, and "1.5B+32B" represents our proposed Speculative Thinking method, where the 32B model supervises reflective reasoning steps of the 1.5B model during inference.

---

[1]Our code is available at https://github.com/uservan/speculative_thinking

# 1   Introduction

Smaller language models are widely used in real-world applications due to their lower computational and memory requirements (Nguyen et al., 2024; Lu et al., 2025; Sui et al., 2025b). However, they often underperform on tasks requiring complex reasoning (Li et al., 2025b; Srivastava et al., 2025; Liu et al., 2025a). Improving their capabilities involves extensive post-training such as supervised fine-tuning on high-quality reasoning traces (Chenglin et al., 2024) or reinforcement learning with verifiable signals (Shao et al., 2024; Chen et al., 2025a; Zhang et al., 2024), which can be costly, data-intensive, and difficult to scale.

To avoid retraining, inference-time scaling methods have been proposed to elicit better intermediate steps from small models (Sui et al., 2025c; Xu et al., 2025). While lightweight and training-free, these approaches depend entirely on the model's existing abilities and often yield limited or inconsistent improvements, particularly on complex tasks Li et al. (2025b). Larger models, by contrast, exhibit significantly stronger reasoning abilities across a wide range of benchmarks (Muennighoff et al., 2025; Ye et al., 2025; Plaat et al., 2024), but their inference cost and latency make them impractical for many deployment scenarios. This tension motivates a central question: *Can we improve small reasoning models during inference by selectively leveraging large models, without additional training?*

Inspired by speculative decoding (Leviathan et al., 2023), which accelerates generation by using a small model to propose tokens later verified by a larger model, we propose **Speculative Thinking**, a training-free framework for improving small-model reasoning during inference. Unlike speculative decoding, which operates at the token level, our approach focuses on reasoning level. A small model generates most of the output but selectively hands off difficult reasoning segments to a stronger model. These segments are identified through structural cues—such as paragraph breaks ("\n\n") followed by reflective phrases like "wait" and "alternatively"—which often mark internal revision. Small models frequently struggle in these cases, producing verbose outputs, while larger models are more concise and effective at backtracking. By dynamically detecting these points and delegating them to a large mentor model, Speculative Thinking preserves the small model's efficiency while leveraging the large model's strength exactly where it matters most.

Empirical results demonstrate the effectiveness of this hybrid approach. A 1.5B model assisted by `Deepseek-distilled Qwen-2.5-32B` improves by +6.6% on AIME, +6.2% on MATH500 (Lightman et al., 2023), +8.1% on GPQA (Rein et al., 2024), and +5.0% on AMC23, while reducing output length—indicating more efficient reasoning. Notably, this approach is also effective for models not explicitly trained for reasoning: `Qwen-2.5-7B-Instruct` gains +7.8% on MATH500 and +14.2% on GPQA when assisted by the 32B mentor.

In summary, Speculative Thinking offers a new inference-time paradigm that fuses the efficiency of small models with the reasoning strength of large models. It opens a promising path toward cost-effective reasoning augmentation for real-world inference.

# 2   Motivations

## 2.1   Analysis of LLM Reasoning Process

This section investigates characteristic patterns that commonly emerge during the reasoning processes of current reasoning models. By analyzing these patterns, we aim to uncover potential avenues for enhancing and optimizing the models' reasoning capabilities.

**"\n\n" acts as a structural clue in model reasoning process.** During inference, reasoning models frequently generate certain reasoning-supportive tokens such as "wait", "hmm" and "alternatively", which are relative with the model's self-reflection behavior. To further analyze them, we examine the preceding token distribution for reasoning-supportive tokens in Deepseek-distilled Qwen-2.5-32B on the MATH500 dataset. As shown in Table 1, we report the top 10 most frequent preceding tokens for three representative reasoning-supportive tokens: "wait", "alternatively", and "hmm". Notably, for all three tokens, the preceding token is overwhelmingly dominated by the newline symbol "\n\n". For instance, in the

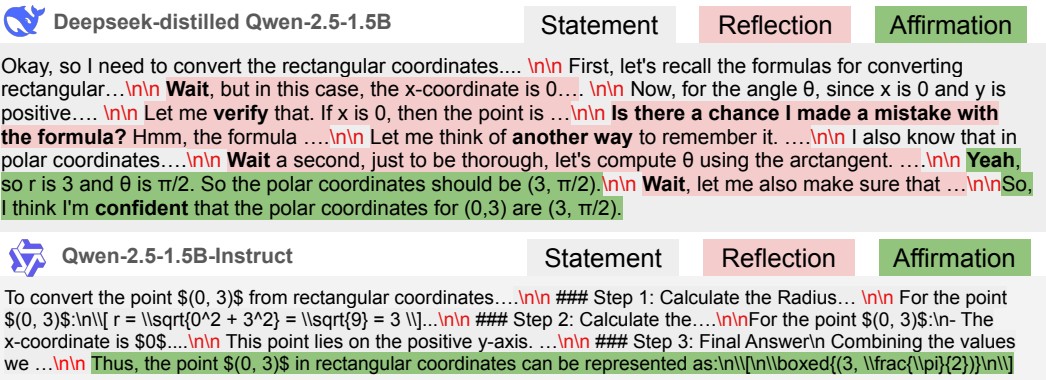

Figure 2: Comparison of outputs between Reasoning Model and Non-reasoning model. Reasoning models often generate negative sentences—typically containing tokens such as *"wait"*—immediately following the delimiter "\n\n". These sentences serve as reflective prompts, helping the model to backtrack, reassess, and verify prior reasoning steps.

Table 1: Proportion of top-10 preceding tokens of reason-supportive words (like wait) in the MATH500 dataset, as generated by the Deepseek-Distilled Qwen-2.5-32B model. We find that over 80% of reasoning-supportive tokens appear after the occurrence of "\n\n", indicating that it plays a crucial role in triggering reflective behavior during reasoning.

| Word | Top 10 frequent tokens before reasoning-supportive tokens (with probability) | | | | |
|---|---|---|---|---|---|
| alternatively | "\n\n" (0.928) | " " (0.050) | ").\n\n" (0.007) | "?\n\n" (0.006) | " \n\n" (0.004) |
| | "].\n\n" (0.002) | "\n\n" (0.001) | ")\n\n" (0.001) | "]\n\n" (0.001) | "?)\n\n" (0.001) |
| hmm | " " (0.690) | ".\n\n" (0.131) | "\n\n" (0.044) | ")\n\n" (0.038) | ").\n\n" (0.035) |
| | "]\n\n" (0.029) | " \n\n" (0.009) | "?\n\n" (0.007) | "?)\n\n" (0.002) | "?"\n\n" (0.002) |
| wait | ".\n\n" (0.699) | " " (0.182) | "?\n\n" (0.039) | ").\n\n" (0.022) | "\n\n" (0.017) |
| | ")\n\n" (0.011) | "]\n\n" (0.007) | " \n\n" (0.007) | ":\n\n" (0.004) | "].\n\n" (0.002) |

case of "wait", over 80% of its preceding tokens are "\n\n". This strongly suggests that *"\n\n" acts as a thinking cue—prompting the model to decide whether to reflect on the previous thought or proceed with the current line of reasoning*. We have also extend this same analysis to other models on the MATH500 dataset in Appendix B.

**Case analysis of LLM reasoning process to prove the role of "\n\n".** To further prove the effect of "\n\n", we conduct a case study on responses generated by Deepseek-distilled Qwen-2.5-1.5B and Qwen-2.5-1.5B-Instruct when answering questions in Figure 2. Specifically, we treat each occurrence of "\n\n" as a delimiter to segment the model's output into multiple parts. We then categorize each segment as **Affirmation**, **Reflection**, or **Statement**: Affirmation segments include affirming expressions such as *yeah* or *yes*, indicating a continuation or endorsement of the preceding thought; Reflection segments contain expressions like *wait*, alternatively, or *hmm*, signaling the model's intent to reflect its previous thought; Statement segments often corresponding to formulaic expressions or factual outputs. Empirical analysis of representative examples in Figure 2 shows that the first sentence after each "\n\n" often contains reasoning-related cues. This suggests that "\n\n" acts as a discourse marker, prompting the model either affirm, reflect or state the previous thought.

## 2.2 Comparisons between Small and Large Reasoning Models

In this section, we compare reasoning models of different sizes to find the differences between small and large reasoning models, including Deepseek-distilled Qwen-2.5-32B, 7B, and 1.5B. Specifically, we analyze their performance differences in terms of accuracy and output length on the AIME 2022-2024 dataset. All the results are shown in Figure 3 and the detailed statistics on other datasets can be found in Appendix G.

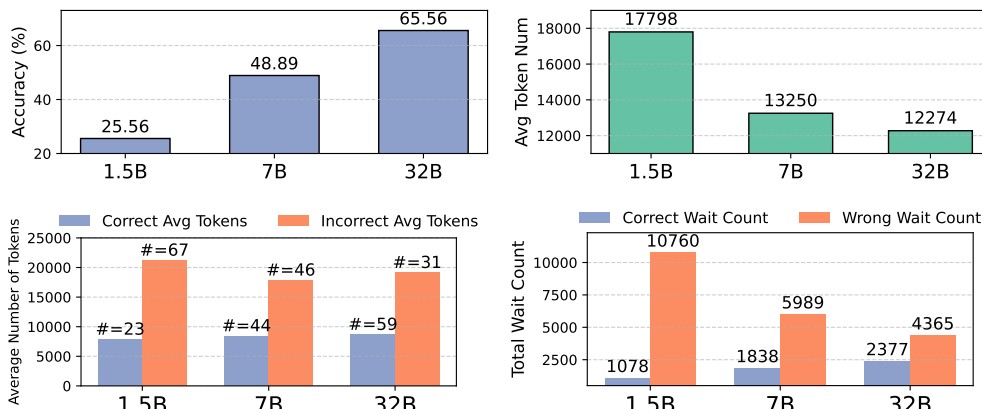

Figure 3: Accuracy and output statistics of three models on the AIME 2022–2024 dataset. Reported metrics include: overall accuracy (upper left), average output length (upper right), average output length (down left) for correct and incorrect answers, as well as the number of reflective sentences—such as those containing terms like "wait" or "alternatively"—in both correct and incorrect responses (down right). "#=67" indicates the number of incorrect responses made by the 1.5B model is 67. The average output length of small models is significantly higher than that of large models. This is primarily due to the excessive length of incorrect responses. At its core, this phenomenon stems from inefficient and redundant self-reflection in small models, which often leads to failed reasoning attempts and ultimately prevents them from arriving at correct answers before its max output length.

**Small reasoning models have worse reasoning performances and much longer responses**. We first report the accuracy and average output length for all three models. As shown in Figure 3, smaller models exhibit significantly lower accuracy compared to larger ones. Interestingly, the average output length of smaller models tends to be much longer. As model size increases, accuracy improves while outputs become more concise. To further understand this phenomenon, we analyze the average lengths of correct and incorrect responses separately. We find that, across all model sizes, incorrect responses are consistently much longer than correct ones. This suggests that the overall average output length is heavily influenced by the proportion of incorrect answers, which are typically more verbose.

**Larger-scale models exhibit more effective self-reflection and backtracking during reasoning.** To further investigate why incorrect responses are substantially longer than correct ones, we analyze the frequency of reflective phrases—such as *"wait"* and *"alternatively"*—which indicate hesitation, self-reflection, or backtracking in reasoning process. As shown in Figure 3, such phrases occur far more frequently in incorrect responses, particularly in smaller models. This suggests that smaller models tend to over-reflect yet under-reason, leading to inefficient exploration of the solution space. Consequently, the excessive length of their outputs is primarily due to their inability to converge on correct answers within the maximum context window, resulting in repetitive branching and redundant verification steps.

## 2.3 How to Combine Small and Large Reasoning Model?

We observe that when reasoning models generate incorrect answers, their average output length increases significantly. A key manifestation of this is the overuse of words like "wait", indicating excessive self-reflection and backtracking. However, as model size increases, such reflection becomes more efficient, resulting in fewer redundant revisions and shorter outputs overall. This naturally raises an intriguing question: **Can the reasoning ability of larger models be leveraged to monitor smaller models during inference?**

We propose a novel intervention strategy that utilizes the "\n\n" reasoning pattern as a control point for collaborative inference. In particular, when a smaller model encounters a "\n\n" followed by tokens like "wait", which often signal confusion or indecision, we can delegate the subsequent reasoning step to a larger model because the larger one could give

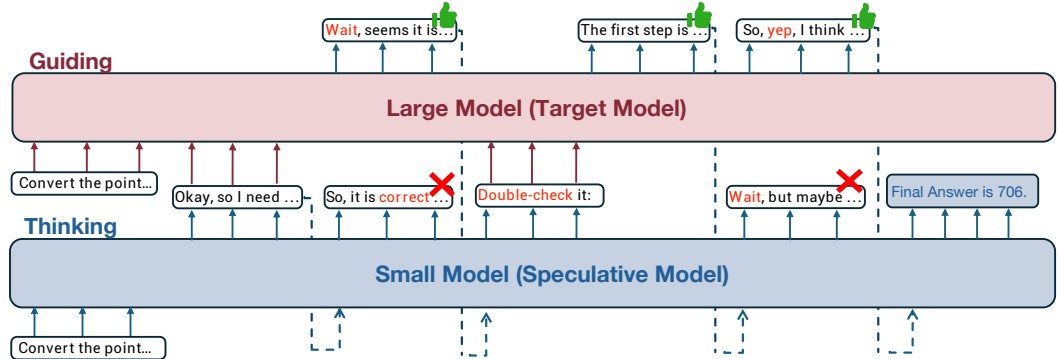

Figure 4: Overview of speculative thinking. A small model generates most output but selectively delegates challenging segments—marked by structural cues such as paragraph breaks ("\n\n") followed by reflective phrases like "wait," "alternatively," or "hold on"—to a stronger model. Small models often produce verbose or incoherent outputs at these points, while larger models handle them concisely. The proposed speculative thinking preserves efficiency while leveraging the large model's strength when most needed.

a more accurate thinking step. The larger model then generates the next thought segment in place of the smaller model, effectively acting as a *reasoning supervisor or corrector*. This large-model-aided intervention may enhance the robustness and accuracy of smaller models by injecting stronger reasoning capabilities, thus balancing efficiency and performance.

## 3    Method: Speculative Thinking

We propose a collaborative inference framework termed **Speculative Thinking**, where a small model acts as *speculative model* and a large model serves as *target model*. Speculative model performs primary reasoning, while target model intervenes selectively to provide auxiliary thoughts when necessary. The overall framework is in Figure 4, . Target model takes over speculative model's generation under the following three scenarios. The hyper-parameters for **Speculative Thinking**—such as the selection of Reflection and Affirmation keywords, and the values of control parameters $n_1$, $n_2$, and $n_3$ are shown in Appendix C.

**(1) Affirmation/Reflection Takeover.** This mechanism leverages stronger reasoning ability of target model to help speculative model decide whether to continue or revise. Speculative model first generates responses until a delimiter token (e.g., \n\n) is encountered. After this delimiter, speculative model generates one full sentence (i.e., $n_1$ tokens). We then classify the sentence into three situations: *Affirmation*, *Reflection*, or *Statement*, based on keyword matching, as shown in Appendix C. If speculative model's sentence is classified as either Affirmation or Reflection, target model immediately takes over and generates $n_1$ tokens. Speculative model then resumes generation conditioned on target model's output.

**(2) Verification Takeover.** We observe that small models often struggle with effective verification. To address this, we introduce a verification-triggered intervention. Whenever a \n\n delimiter is encountered—regardless of whether the subsequent sentence is generated by the speculative or target model—we examine if the sentence contains verification-related cues (e.g., *verify*, *double-check*, etc.). If such cues are detected, target model takes over to generate $n_2$ tokens, assisting the verification process and mitigating false conclusions.

**(3) Excessive Reflection Takeover.** Our analysis reveals that a hallmark of incorrect answers is excessive backtracking, where the model repeatedly negates its own thoughts. To mitigate this, we implement a negativity counter $c$ that tracks the number of reflection sentences. Each time a \n\n is encountered, we evaluate whether the following sentence is negative; if so, we increment $c$. Once $c$ exceeds a predefined threshold, we prompt the model to exit the reflection loop. Specifically, we insert an auxiliary sentence (e.g., *"Let us check whether there are some wrong steps."*) into the output, and then delegate the next $n_3$ tokens to target model. This mechanism serves to reorient speculative model and prevent reflection thinking loops.

Table 2: Accuracy, average output length, and estimated speed of models on four datasets. Here, 1.5B refers to the Deepseek-Distilled Qwen-2.5-1.5B model. "+" means with the help of large models. modify ratio indicates the proportion of tokens in the final output that come from the target model. After applying Speculative Thinking, both 1.5B and 7B models demonstrate improvements in accuracy, output length, and estimated inference speed. The improvement in estimated speed is measured relative to the corresponding target model.

| Dataset pass@1 | Speculative Model | Target Model | Modify Ratio | Acc (%) | Improv. | Length Avg | Decr. | Estimated Speed | Improv. |
|---|---|---|---|---|---|---|---|---|---|
| AIME | 1.5B | − | − | 25.6 | − | 17800.0 | − | 198.9 | − |
| | | +14B | 18.0% | 33.3 | +7.7 | 16691.2 | -6.2% | 110.3 | +121.1% |
| | | +32B | 19.0% | 32.2 | +6.6 | 15706.1 | -11.7% | 85.8 | +185.9% |
| | 7B | − | − | 48.9 | − | 13250.4 | − | 56.4 | − |
| | | +32B | 18.0% | 53.3 | +4.4 | 13213.6 | -0.3% | 41.0 | +36.8% |
| | 14B | − | − | 60.0 | − | 12600.2 | − | 49.9 | − |
| | 32B | − | − | 65.6 | − | 12274.3 | − | 30.0 | − |
| GPQA | 1.5B | − | − | 33.8 | − | 7922.0 | − | 223.2 | − |
| | | +14B | 15.0% | 38.9 | +5.1 | 8134.3 | +2.7% | 128.1 | +121.7% |
| | | +32B | 17.0% | 41.9 | +8.1 | 7612.4 | -3.9% | 91.8 | +190.4% |
| | 7B | − | − | 45.5 | − | 6111.5 | − | 62.1 | − |
| | | +32B | 22.0% | 52.0 | +6.5 | 5952.5 | -2.6% | 40.3 | +27.5% |
| | 14B | − | − | 57.1 | − | 5762.7 | − | 57.8 | − |
| | 32B | − | − | 61.6 | − | 5406.8 | − | 31.6 | − |
| MATH500 | 1.5B | − | − | 83.2 | − | 5439.1 | − | 242.6 | − |
| | | +14B | 19.0% | 89.0 | +5.8 | 4527.4 | -16.8% | 134.6 | +124.0% |
| | | +32B | 19.0% | 89.4 | +6.2 | 4582.8 | -15.7% | 96.6 | +200.0% |
| | 7B | − | − | 92.8 | − | 3975.2 | − | 63.7 | − |
| | | +32B | 18.0% | 93.0 | +0.2 | 3767.8 | -5.2% | 46.0 | +42.9% |
| | 14B | − | − | 93.8 | − | 3609.0 | − | 60.1 | − |
| | 32B | − | − | 92.8 | − | 3802.2 | − | 32.2 | − |
| AMC23 | 1.5B | − | − | 75.0 | − | 10460.8 | − | 212.7 | − |
| | | +14B | 19.0% | 85.0 | +10.0 | 7503.2 | -28.3% | 123.7 | +123.0% |
| | | +32B | 21.0% | 80.0 | +5.0 | 8691.2 | -16.9% | 82.8 | +170.0% |
| | 7B | − | − | 92.5 | − | 6093.8 | − | 62.6 | − |
| | | +32B | 16.0% | 92.5 | +0.0 | 5116.1 | -16.1% | 48.0 | +56.4% |
| | 14B | − | − | 95.0 | − | 6395.4 | − | 55.5 | − |
| | 32B | − | − | 95.0 | − | 7106.7 | − | 30.7 | − |

# 4 Experiments

## 4.1 Large Reasoning Models Monitor Small Reasoning Models

This experiment aims to evaluate the effectiveness of Speculative Thinking. We adopt three key evaluation metrics: **accuracy**, **average output length**, and **estimated inference speed**, to fully assess the trade-off between reasoning performance and efficiency. The rationale for choosing the estimated inference speed, along with the details of its computation, is provided at the end of this section. We conduct experiments on four benchmark datasets: AIME 2022–2024, GPQA-Diamond, MATH500, and AMC23.

**Analysis of results of Large Reasoning Models Monitor Small Reasoning Models.** The results are summarized in Table 2, which demonstrates that our method consistently improves accuracy while reducing unnecessary output length and enhancing inference speed. For example, after being assisted by the 32B target model, the 1.5B speculative model demonstrates consistent and significant improvements across multiple datasets. Specifically, its accuracy increases by **6.2%** on MATH500, **8.1%** on GPQA, **5.0%** on AMC23, and **6.6%** on AIME. In addition, the average output length is reduced by **15.7%**, **3.9%**, **16.9%** and **11.7%** on the same datasets, respectively, indicating that the speculative model is able to reach conclusions more efficiently with guidance from the large model. Furthermore, in terms of estimated

generation speed, the 1.5B model assisted by the 32B model consistently outperforms the standalone 32B model, despite leveraging it selectively. These findings collectively demonstrate the effectiveness and practicality of our *Speculative Thinking* framework, offering a promising trade-off between performance and computational efficiency. Moreover, when assisting the smaller reasoning model, the target model only needs to modify approximately **20%** of the speculative model's output to significantly enhance its reasoning performance.

**Theoretical Estimation of FLOPs and Token Generation Speed**. We adopt a theoretical analysis rather than empirical timing, since our method—*Speculative Thinking*—primarily introduces logical coordination between models. In contrast, runtime measurements would be significantly affected by backend GPU optimizations, especially in systems like vLLM ([Kwon et al., 2023](#)). The computation of FLOPs for prefill and decode stages is in [Appendix A](#). The differences between prefix and decode are shown in [Figure 5](#).

Figure 5: A comparison between the prefix and decode stages reveals that the time (in seconds) required to process multiple tokens during the prefix phase is nearly equivalent to the time taken to decode a single token.

| Model | decode | prefix | | |
|-------|--------|--------|------|------|
| | n=1 | n=1 | n=20 | n=250 |
| 1.5B | 0.036 | 0.036 | 0.040 | 0.045 |
| 32B | 0.09 | 0.11 | 0.12 | 0.15 |

We empirically profile average inference time for both decode and prefix stages across various model sizes and output token lengths. These measurements are obtained using `generate()` api from HuggingFace Transformers, with key-value cache enabled for the prompt. We observe that when GPU memory are sufficient, the average time in prefix stage remains relatively stable across positions. We could see time required to process multiple tokens during the prefix phase is nearly equivalent to the time taken to decode a single token. To reflect the difference, we assume a speedup for the prefix stage : $\text{FLOPs}_{\text{prefix}}(m) = \text{FLOPs}_{\text{decode}}(n = 1)$, where m and n mean the token number. We set GPU computational capacity to $3.12 \times 10^{10}$ FLOPs/s, which corresponds to a A100-class GPU. The estimated speed is calculated as follows:

$$\text{Estimated Speed} = \frac{\text{Total Tokens}}{\left(\text{FLOPs}_{\text{prefill}} + \text{FLOPs}_{\text{prefix}} + \text{FLOPs}_{\text{decode}}\right)/\text{GPU Capacity}} \quad (1)$$

### 4.2 Reasoning Models Monitor Non-Reasoning Models

Given that large reasoning models can effectively assist smaller reasoning models, a natural follow-up question is: *Can we leverage reasoning-capable models to enhance the performance and accuracy of non-reasoning models*? To explore this, we adapt the *Speculative Thinking* framework to monitor a speculative model that lacks inherent reasoning capability.

**Modification for speculative thinking applied to non-reasoning models**. Specifically, in Affirmation/Reflection Takeover, we originally determine whether the speculative model's sentence following a "\n\n" contains reflective or Affirmative reasoning cues. However, non-reasoning models typically do not emit such linguistic signals. Therefore, in this setting, we directly allow target model to take over and generate the next sentence after each "\n\n" . In addition, we further enhance the speculative model by allowing target model to generate the first 100 tokens before any question answering begins. This is motivated by the observation that reasoning models often preface their answers with structured setups such as *"Okay, so I have this problem where I need..."*, which helps guide the generation for models.

**Analysis of Results of Reasoning Models Monitor Non-Reasoning Models.** The results, where a non-reasoning model is augmented by a reasoning-capable target model, are shown in [Table 3](#). We first observe that `Qwen-2.5-7B-Instruct`, a non-reasoning model, benefits notably from speculative assistance by both 7B and 32B reasoning models. For instance, on the MATH500 dataset, its accuracy improves from 74.0% to 81.8%. However, this improvement comes at the cost of increased output length, indicating a trade-off between enhanced reasoning ability and generation efficiency. However, when assisted by the 1.5B reasoning model, performance improvements are not consistently observed. This indicates

Table 3: Accuracy, average output length, and estimated speed on four datasets. 7B-Instruct refers to Qwen-2.5-7B-Instruct. "+" means with the help of reasoning models. Modify ratio indicates the proportion of tokens in the final output that come from target model. After applying Speculative Thinking, models demonstrate improvements in accuracy. The improvement in estimated speed is measured relative to the corresponding target model.

| Dataset pass@1 | Speculative Model | Target Model | Avg Length | Modify Ratio | Estimated Speed | Acc (%) | Acc Improv. |
|---|---|---|---|---|---|---|---|
| AIME | 7B-Instruct | – | 1249.8 | – | 64.7 | 7.8 | – |
| | | +1.5B | 8029.3 | 54.0% | 51.5 | 6.7 | -1.1 |
| | | +7B | 10458.5 | 42.0% | 38.8 | 13.3 | +5.5 |
| | | +32B | 10236.0 | 46.0% | 29.0 | 15.6 | +7.8 |
| GPQA | 7B-Instruct | – | 5.6 | – | 1.5 | 33.8 | – |
| | | +1.5B | 6763.8 | 43.0% | 45.6 | 31.8 | -2.0 |
| | | +7B | 4739.7 | 42.0% | 36.8 | 40.9 | +7.1 |
| | | +32B | 6652.8 | 31.0% | 33.6 | 48.0 | +14.2 |
| MATH500 | 7B-Instruct | – | 802.3 | – | 58.3 | 74.0 | – |
| | | +1.5B | 3368.8 | 43.0% | 53.1 | 74.8 | +0.8 |
| | | +7B | 3172.0 | 44.0% | 41.2 | 79.2 | +5.2 |
| | | +32B | 3015.9 | 44.0% | 31.7 | 81.8 | +7.8 |
| AMC23 | 7B-Instruct | – | 878.5 | – | 64.8 | 42.5 | – |
| | | +1.5B | 7603.0 | 49.0% | 48.4 | 55.0 | +12.5 |
| | | +7B | 6431.5 | 43.0% | 39.0 | 67.5 | +25.0 |
| | | +32B | 8732.8 | 31.0% | 33.5 | 55.0 | +12.5 |

that, during the design of speculative thinking systems, it is preferable to choose a target model that is either of equal size or larger than the speculative model, and more importantly, possesses stronger reasoning capabilities. Mismatches where the speculative model is larger or stronger than the target model may lead to suboptimal or even detrimental outcomes.

### 4.3 Speculative Thinking Generalize to Different Families of Models

Speculative thinking is not limited to a specific model family; rather, it generalizes well across diverse reasoning models. In this section, we validate this generalization ability through two types of evidence: (1) prefix pattern analysis across multiple model families, and (2) the performances on cross-family speculative-target settings.

We first examine whether the "\n\n + wait" pattern is model-specific or broadly adopted. We analyze three strong open-source reasoning models—QwQ-32B (Qwen), Phi-4-reasoning-plus (Microsoft), and AceReason-Nemotron-14B (NVIDIA)—on the MATH500 benchmark. Specifically, we collect the top-10 tokens immediately preceding the token "wait", which often marks the onset of a reasoning trace. As shown in Table 4, models frequently precede "wait" with structural tokens like "\n\n", suggesting that multi-line breaks act as natural separators for reasoning stages. This supports the hypothesis that speculative thinking can identify reasoning boundaries based on structural cues shared across model families.

To further test generalization across model families, we evaluate speculative thinking in two heterogeneous settings: (1) a small Qwen model guiding a Phi model, and (2) a small LLaMA3 model guiding a large Qwen model. Both pairs differ not only in scale but also in underlying architecture. Table 5 presents results on MATH500. In both cases, speculative thinking significantly improves performance over the smaller model alone, while achieving notable speedups compared to the larger model. This demonstrates that speculative thinking remains effective even when the speculative and target models belong to different families.

### 4.4 Comparisons between Speculative Decoding and Speculative Thinking

This experiment primarily compares the differences between speculative decoding and speculative thinking. Due to the constraint that speculative decoding requires the speculative model and the target model to have the same vocabulary size, we obtain speculative

Table 4: Top-10 most frequent tokens preceding the `wait` token across different reasoning models on the MATH500 benchmark. Structural tokens like "\n\n" and ".\n\n" dominate across models, highlighting their role in segmenting and triggering reasoning behaviors.

| Model | Top 10 frequent tokens before `wait` | | | | |
|---|---|---|---|---|---|
| QwQ-32B (Qwen) | ".\n\n" | " " | "?\n\n" | " \n\n" | ":\n\n" |
| | "\n\n" | ",\n\n" | ").\n\n" | ")\n\n" | " ?\n\n" |
| Phi-4-reasoning-plus | " " | ".\n\n" | ").\n\n" | " \n\n" | " \n" |
| | "\n\n" | ""\n\n" | ")\n\n" | ".\n" | ",...\n\n" |
| AceReason-Nemotron-14B | ".\n\n" | " " | "?\n\n" | "\n\n" | """ |
| | " \n\n" | ").\n\n" | ":\n\n" | ")\n\n" | "]\n\n" |

Table 5: Accuracy, average output length, and estimated speed on MATH500 for different speculative-target model pairs. "Small" refers to the speculative model, "Large" refers to the target model. "LLaMA3-8B" denotes `DeepSeek-R1-Distill-LLaMA3-8B`, "Qwen-32B" denotes `DeepSeek-R1-Distill-Qwen-32B`, and "Phi-4" denotes `Phi-4-reasoning-plus`.

| Small Model | Large Model | Avg Length | Speed | Speedup (%) | Acc (%) | Improv. |
|---|---|---|---|---|---|---|
| LLaMA3-8B | – | 4513.5 | 69.7 | – | 85.8 | – |
| LLaMA3-8B | Qwen-32B | 4082.2 | 46.5 | +44.4% | 89.2 | +3.4 |
| Qwen-32B | – | 3802.1 | 32.2 | – | 92.8 | – |
| Qwen-1.5B | – | 5439.1 | 242.6 | – | 83.2 | – |
| Qwen-1.5B | Phi-4 | 3768.9 | 98.0 | +160.6% | 90.0 | +6.8 |
| Phi-4 | – | 6873.7 | 37.6 | – | 93.4 | – |

decoding results where the speculative model is 7B, and the target model is 32B. To align with Speculative Thinking, which takes over the generation of 20 tokens at a time, we set the speculative model in speculative decoding to generate n = 20 tokens per step.

Speculative decoding relies on the speculative and target models having similar token output distributions to accelerate generation. In contrast, Speculative Thinking focuses on enhancing the speculative model's reasoning with lightweight assistance from target model, without strictly requiring token distributional alignment. As shown in in Figure 6, although speculative decoding matches the accuracy of 32B model, it often suffers from a high rejection rate—nearly 50% of tokens need to be regenerated by target model, which diminishes its speed. Speculative Thinking avoids this issue by allowing the target model to intervene only when necessary, improving the speculative model's reasoning with minimal overhead.

## 5 Related Works

**LLM Reasoning.** Current approaches to enhancing the reasoning capabilities (Chen et al., 2025a; Plaat et al., 2024; Sun et al., 2023) of language models primarily fall into two categories: reinforcement learning (Schulman et al., 2017) and supervised fine-tuning (Jaech et al., 2024; Yang et al., 2024). For instance, DeepSeek (Guo et al., 2025; Liu et al., 2024) achieved state-of-the-art reasoning performance using GRPO (Shao et al., 2024; Yu et al., 2025), and further improved smaller models by distilling high-quality reasoning traces. This line of research has inspired numerous efforts to replicate DeepSeek-R1 with the goal of uncovering potential "aha moments" in reasoning, including works such as Logic RL (Xie et al., 2025) and SimpleRL-Zoo (Zeng et al., 2025). Many studies also use SFT to improve reasoning, including SkyThought-T1 (Team, 2025b) and Bespoke-Stratos-32B (Labs, 2025), which collect and fine-tune on carefully curated high-quality reasoning data. Several works have further investigated key techniques for enhancing reasoning performance during RL (Baek & Tegmark, 2025; Yeo et al., 2025) or SFT (Chen et al., 2025b; 2024a; Tian et al., 2025; Liu et al., 2025b). For example, (Li et al., 2025a) argues that the structure of reasoning steps in the data is more critical than the actual content; (Ji et al., 2025) highlights the importance of the

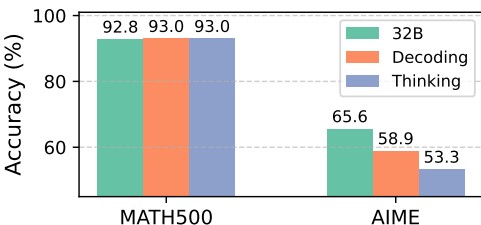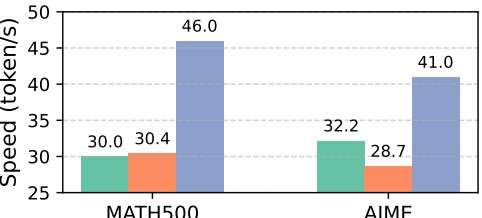

Figure 6: Comparison between Speculative Decoding and Thinking using a 7B speculative model and a 32B target model. In Speculative Decoding, speculative model generates 20 tokens per step to match the number of intervention tokens in Speculative Thinking.

initial few tokens in each reasoning instance for optimizing model performance. In addition, several recent studies—such as s1(Muennighoff et al., 2025) emphasize the value of selecting a small set of high-quality reasoning samples to drive efficient model improvement.

**Efficient Reasoning.** Current reasoning models still exhibit notable limitations (Bandyopadhyay et al., 2025; Li et al., 2025c). One prominent issue is excessive response length—many reasoning-enabled models tend to generate unnecessarily verbose outputs. As a result, efficient reasoning has become an emerging research focus. An early effort in this direction was proposed by Kimi 1.5 (Team et al., 2025), which introduced the Long-to-Short method. This approach collects paired long and short responses and applies Direct Preference Optimization (Rafailov et al., 2023; Zeng et al., 2024) to train models that prefer concise answers. The idea was later reproduced by Sky-Thought (Team, 2025a), further validating its effectiveness. TokenSkip (Xia et al., 2025), which improves efficiency by identifying and removing redundant or uninformative tokens to create cleaner training data. LightThinker (Zhang et al., 2025) takes a different route by explicitly compressing intermediate thoughts to generate shorter yet informative reasoning traces, thereby enabling models to produce more concise outputs via fine-tuning. Wang et al. (2025); Sui et al. (2025a) highlights a counterintuitive phenomenon: when reasoning fails, model outputs often become significantly longer. This is attributed to repetitive generation of reasoning-supportive tokens like "wait", which reflect the model's tendency to over-compensate by generating more thoughts. Other notable approaches include Dynasor(Fu et al., 2024), which uses probing techniques to detect and terminate reasoning early. There are some other works including efficient reaosning (Aytes et al., 2025; Lee et al., 2025; Sui et al., 2025c; Xu et al., 2025; Liao et al., 2025).

## 6 Conclusion

We propose Speculative Thinking, a training-free framework that leverages larger reasoning models to guide smaller ones through selective delegation at structurally meaningful points in generation. By exploiting the natural reasoning patterns of LLMs—particularly reflection cues like "\n\n"—our approach significantly enhances both accuracy, average output length and efficiency without any additional training in four math reasoning datasets like MATH500. Experiments demonstrate substantial gains in performance and output conciseness, underscoring the potential of collaborative inference between models of different capacities. This highlights a promising paradigm for improving reasoning of reasoning and non-reasoning models without additional data or training computation cost.

## Limitations

First, the method depends on specific structural cues (e.g., \n\n followed by keywords), whose stability across different model sizes, families, and training setups remains uncertain. Second, the approach involves several hyperparameters (e.g., cue keywords, token span thresholds), which may require careful tuning for different tasks and model pairs.

## Acknowledgments

This research was partially supported by NSF Awards OAC-2117439. Further, this work made use of the High Performance Computing Resource in the Core Facility for Advanced Research Computing at Case Western Reserve University (CWRU). We give special thanks to the CWRU HPC team for their prompt and professional help and maintenance. The views and conclusions in this paper are those of the authors and do not represent the views of any funding or supporting agencies.

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

# A Compuation of FLOPs

$$\text{FLOPs}_{\text{prefill}}(s) = 8sh^2 + 16sh + 4s^2h + 4s^2n + 6shh' + 2sh' \tag{2}$$

$$\text{FLOPs}_{\text{decode}}(s) = 8h^2 + 16h + 4sh + 4sn + 6hh' + 2h' \tag{3}$$

$$\text{FLOPs}_{\text{total}} = \text{FLOPs}_{\text{prefill}}(p_l) + \sum_{i=0}^{d_l-1} \text{FLOPs}_{\text{decode}}(p_l + i) \tag{4}$$

We compute the FLOPs of prefill and decoding stages based on Chen et al. (2024b); Han (2024), where the batch size is 1. $s$ is the input sequence length. $h$ is the hidden size. $h'$ is the intermediate size of the feed-forward network (FFN). $n$ is the number of attention heads. $d$ is the size of each attention head, such that $h = nd$. $p_l$ is the length of the problem prompt. $d_l$ is the number of tokens to be generated in the solution.

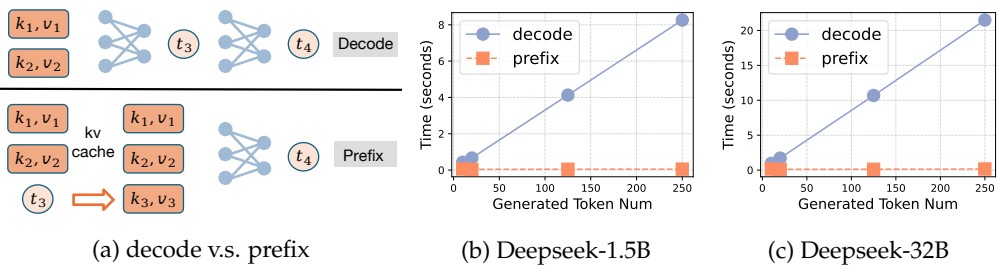

(a) decode v.s. prefix    (b) Deepseek-1.5B    (c) Deepseek-32B

Figure 7: Comparison between Decode and Prefix stages: average time consumed by the 1.5B and 32B models when generating different numbers of output tokens. As the number increases, decoding time grows significantly, while prefix time remains nearly constant.

# B Proportion of Top-10 Preceding Tokens

Table 6: Proportion of top-10 preceding tokens of reason-supportive words (like wait) in the MATH500 dataset, as generated by the Deepseek-Distilled Qwen-2.5-1.5B model.

| Word | Top 10 frequent tokens before reasoning-supportive tokens (with probability) | | | | |
|---|---|---|---|---|---|
| alternatively | "\n\n" (0.708) | " " (0.207) | " " (0.055) | ").\n\n" (0.011) | "?\n\n" (0.008) |
| | " \n\n" (0.004) | "\n\n" (0.003) | ")\n\n" (0.001) | ":\n\n" (0.001) | " )\n\n" (0.001) |
| hmm | " " (0.689) | ".\n\n" (0.139) | ")\n\n" (0.043) | "]\n\n" (0.037) | "\n\n" (0.033) |
| | ").\n\n" (0.027) | " " (0.007) | "]\n" (0.007) | "?\n\n" (0.004) | " \n\n" (0.004) |
| wait | ".\n\n" (0.647) | " " (0.230) | "?\n\n" (0.044) | ").\n\n" (0.026) | "\n\n" (0.016) |
| | ")\n\n" (0.009) | "]\n\n" (0.007) | " \n\n" (0.005) | " " (0.004) | ":\n\n" (0.002) |

Table 7: Proportion of top-10 preceding tokens of reason-supportive words (like wait) in the MATH500 dataset, as generated by the Deepseek-Distilled Qwen-2.5-7B model.

| Word | Top 10 frequent tokens before reasoning-supportive tokens (with probability) | | | | |
|---|---|---|---|---|---|
| alternatively | "\n\n" (0.929) | " " (0.048) | "?\n\n" (0.008) | ").\n\n" (0.007) | " \n\n" (0.004) |
| | ")\n\n" (0.001) | "?)\n\n" (0.001) | "].\n\n" (0.000) | "]\n\n" (0.000) | "'.\n\n" (0.000) |
| hmm | " " (0.697) | ".\n\n" (0.123) | "\n\n" (0.047) | ")\n\n" (0.043) | "]\n\n" (0.038) |
| | ").\n\n" (0.025) | "?\n\n" (0.006) | " \n\n" (0.005) | "]\n" (0.003) | " )\n\n" (0.003) |
| wait | ".\n\n" (0.637) | " " (0.224) | "?\n\n" (0.048) | ").\n\n" (0.029) | "\n\n" (0.019) |
| | ")\n\n" (0.015) | " \n\n" (0.007) | "]\n\n" (0.005) | ":\n\n" (0.004) | " )\n\n" (0.002) |

Table 8: Proportion of top-10 preceding tokens of reason-supportive words (like wait) in the MATH500 dataset, as generated by the Deepseek-Distilled Qwen-2.5-14B model.

| Word | Top 10 frequent tokens before reasoning-supportive tokens (with probability) | | | | |
|---|---|---|---|---|---|
| alternatively | "\n\n" (0.867) | " " (0.076) | ").\n\n" (0.022) | "?\n\n" (0.015) | " \n\n" (0.013) |
| | ")\n\n" (0.001) | "\n\n" (0.001) | "]\n\n" (0.001) | "].\n\n" (0.001) | "   " (0.001) |
| hmm | "   " (0.649) | ".\n\n" (0.159) | "\n\n" (0.047) | ")\n\n" (0.036) | "]\n\n" (0.033) |
| | ").\n\n" (0.033) | " \n\n" (0.010) | "?\n\n" (0.009) | "]\n" (0.007) | }\n\n (0.004) |
| wait | ".\n\n" (0.643) | "   " (0.206) | "?\n\n" (0.053) | ").\n\n" (0.032) | "\n\n" (0.021) |
| | " \n\n" (0.015) | ")\n\n" (0.013) | "]\n\n" (0.004) | ":\n\n" (0.003) | "?)\n\n" (0.001) |

## C  Hyperparameters of Speculative Thinking

A sentence is labeled Affirmation or Reflection if it contains affirmation cues (e.g., *yes, yep*) or backtracking cues (e.g., *wait, alternatively*); and Statement if neither type is present. If both Affirmation and Reflection keywords appear, the decision is made based on majority count, and in case of a tie, we default to Reflection.

Within the proposed framework, we define three sets of indicative keywords that trigger different forms of target model intervention:

- **Reflection keywords**, used to detect reflection or hesitation: *"wait"*, *"alternatively"*, *"hold on"*, *"another"*, *"verify"*, *"think again"*, *"recap"*, *"check"*.

- **Affirmation keywords**, indicating confidence or commitment to a line of reasoning: *"yeah"*, *"yes"*, *"final answer"*, *"confident"*.

- **Verification keywords**, used to trigger verification-based intervention: *"verify"*, *"think again"*, *"recap"*, *"check"*.

We also configure fixed token lengths for the target model's interventions in different scenarios: $n_1 = 20$ for Affirmation/Reflection Takeover, $n_2 = 125$ for Verification Takeover, and $n_3 = 125$ for Excessive Negativity Takeover. These hyperparameters are selected to balance informativeness and computational cost.

## D  Hyperparameter Choices and Ablation Analyses

Speculative Thinking requires several key hyperparameters to be selected carefully for different model pairs and tasks. In this appendix, we provide implementation details, empirical justifications, and ablation results for these hyperparameters.

### D.1  Classification Keyword Selection

Speculative Thinking relies on detecting semantic cues such as Affirmation, Reflection, and Verification. These cues are used to classify generated segments and guide speculative decisions.

We first analyze model generations on reasoning datasets and extract leading tokens from the first sentence after the occurrence of \n\n. We then rank their frequency and group representative ones into three keyword categories.

To validate their importance, we conduct an ablation study using reduced keyword sets (one keyword per category):

- Affirmation: "confident"

- Reflection: "wait"

- Verification: "verify"

Table 9: Ablation on classification keyword set size. Reduced keyword sets hurt performance.

| Setting | MATH500 Acc. | AIME Acc. |
|---|---|---|
| Full keyword set | 89.4 | 32.2 |
| Reduced (1 per category) | 87.8 | 26.67 |

This result highlights the benefit of using a richer lexical cue set. While training-based classification is possible, we retain our lightweight heuristic approach to preserve generality and efficiency.

### D.2 Negativity Counter Threshold

The negativity counter threshold $c$ controls early termination during excessive speculative reflection. We set $c = 15$ based on empirical observation and fix it across all experiments for simplicity.

We test three values: low ($c = 3$), default ($c = 15$), and high ($c = 50$):

Table 10: Effect of negativity threshold $c$ on performance. $c = 15$ offers a good trade-off.

| Setting | MATH500 Acc. | Speed (tok/s) |
|---|---|---|
| $c = 3$ | 0.890 | 80.25 |
| $c = 15$ (default) | 0.894 | 96.60 |
| $c = 50$ | 0.876 | 101.17 |

The results indicate that overly low thresholds harm speed, while overly high ones hurt accuracy. A fixed $c = 15$ balances both well, though adaptive tuning remains a promising direction.

### D.3 Importance of Initial Target Generation

When speculative thinking involves a non-reasoning speculative model and a reasoning-capable target model, the initial generation from the target model plays a critical role in alignment.

We compare setups with and without the initial 100-token generation from the target:

Table 11: Ablation on the presence of initial target generation. Skipping it leads to large performance drops.

| Setting | MATH500 Acc. | AIME Acc. |
|---|---|---|
| 7B + 7B (w/o init gen) | 73.8 | 6.67 |
| 7B + 7B (with init gen) | 79.2 | 13.3 |
| 7B + 32B (w/o init gen) | 74.2 | 6.67 |
| 7B + 32B (with init gen) | 81.8 | 15.6 |

These results show that the initial generation provides necessary scaffolding for the reasoning model to align effectively.

## E   Results of Non-reasoning model

We observe that speculative thinking improves the accuracy of the 1B-Instruct model across most datasets, with particularly large gains on AIME and GPQA. While performance on MATH500 slightly drops, the overall trade-off remains favorable in terms of accuracy and

speed. These results confirm that even non-reasoning models can benefit from speculative alignment with stronger reasoning models.

Table 12: Accuracy, average output length, and estimated speed on four datasets. 1B-Instruct refers to Qwen-2.5-1.5B. "+" means with the help of reasoning models. Modify ratio indicates the proportion of tokens in the final output that come from target model. After applying Speculative Thinking, 1B-Instruct models demonstrate improvements in accuracy

| dataset pass@1 | speculative model | target model | avg length | modify ratio | estimated speed | acc (%) | Improv. |
|---|---|---|---|---|---|---|---|
| AIME | 1B-Instruct | normal | 1701.5 | – | 224.4 | 4.4 | – |
| | | +7B | 14240.7 | 37.0% | 76.9 | 8.9 | +102.3% |
| | | +32B | 15536.7 | 34.0% | 51.6 | 10.0 | +127.3% |
| GPQA | 1B-Instruct | normal | 694.9 | – | 164.9 | 23.7 | – |
| | | +7B | 9019.3 | 26.0% | 95.4 | 30.3 | +27.8% |
| | | +32B | 10500.2 | 26.0% | 62.4 | 33.3 | +40.5% |
| MATH500 | 1B-Instruct | normal | 1424.1 | – | 205.4 | 50.2 | – |
| | | +7B | 7947.2 | 30.0% | 58.7 | 48.8 | -2.9% |
| | | +32B | 8935.7 | 29.0% | 89.7 | 48.2 | -4.0% |
| AMC23 | 1B-Instruct | normal | 1605.0 | – | 217.6 | 20.0 | – |
| | | +7B | 19376.5 | 23.0% | 89.2 | 27.5 | +37.5% |
| | | +32B | 17114.4 | 23.0% | 65.4 | 30.0 | +50.0% |

# F Results of Deepseek-Distilled Qwen-2.5-7B

We present the accuracy and average output length of Deepseek-Distilled Qwen-2.5-7B on four datasets.

Across all four datasets, Speculative Thinking consistently improves the accuracy of the 7B model while reducing its average output length. This indicates that the 32B model effectively guides the 7B model toward more concise and accurate responses. The improvements are especially pronounced on complex benchmarks like MATH500 and AMC23, highlighting the benefits of reasoning-aware alignment.

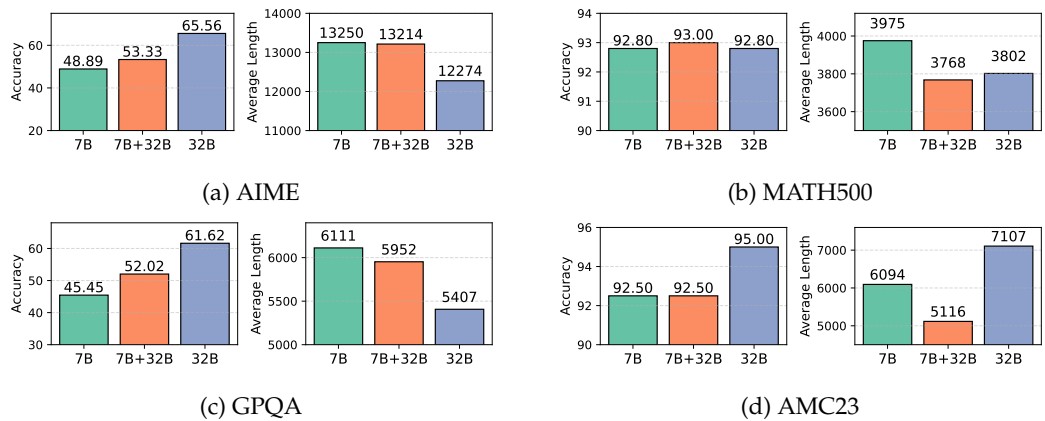

(a) AIME  (b) MATH500

(c) GPQA  (d) AMC23

Figure 8: Accuracy and average output length of models on four datasets (AIME 2020–2024, MATH500, GPQA, and AMC23). 1B denotes Deepseek-Distilled Qwen 2.5-7B model, 32B refers to Deepseek-Distilled Qwen 2.5-32B model, and 7B+32B represents Speculative Thinking, where 32B model assists 7B model. Speculative Thinking leads to a significant improvement in the 7B model's accuracy while effectively reducing its output length.

# G Statistics of Different Size model

Overall, we observe consistent trends across datasets: larger models produce more accurate answers, longer outputs, and higher reflection token frequencies, indicating deeper reasoning. In contrast, smaller models tend to generate shorter, more direct responses with fewer reasoning cues. These patterns support our design of speculative thinking, where larger models act as reasoning specialists, and smaller models provide faster but less reflective baselines.

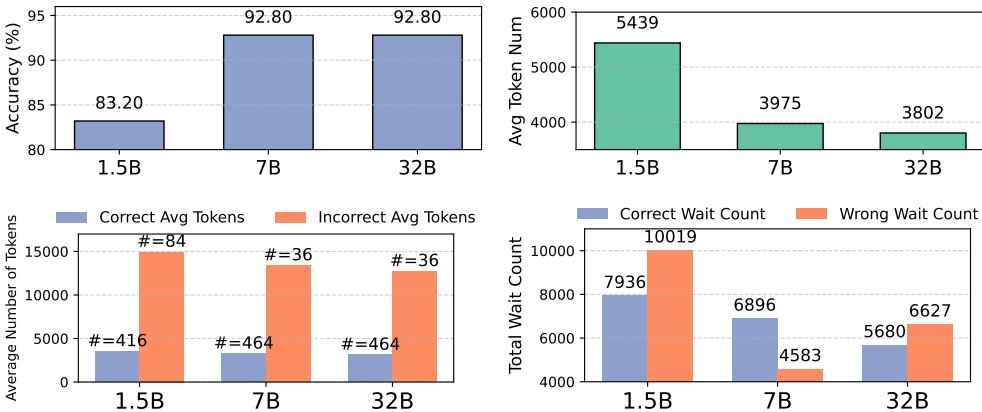

Figure 9: Accuracy and output statistics of three models on the MATH500 dataset.

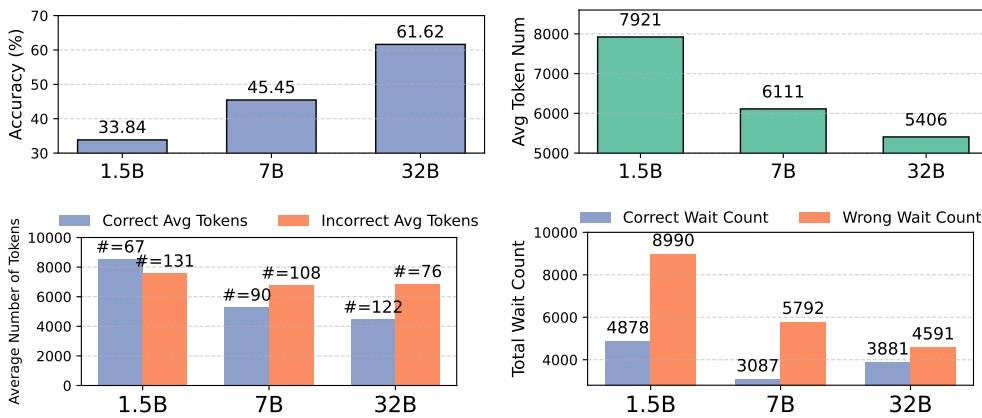

Figure 10: Accuracy and output statistics of three models on the GPQA dataset.

# H Application Scenarios of Speculative Thinking

Speculative Thinking is particularly well-suited for scenarios where efficiency, cost, and scalability are prioritized, and some accuracy trade-off is acceptable. Below we outline representative deployment contexts where this approach is not only reasonable, but often desirable:

- **Educational QA Platforms.** When serving thousands of students concurrently, small models can quickly handle routine questions, while difficult queries invoke speculative assistance from a 32B reasoning model—balancing cost and accuracy.

- **Enterprise Assistants and Internal Copilots.** Productivity systems often require low-latency responses and reasonable cost control. Here, speculative thinking allows lightweight models to dominate, while selectively involving stronger models for more complex requests.

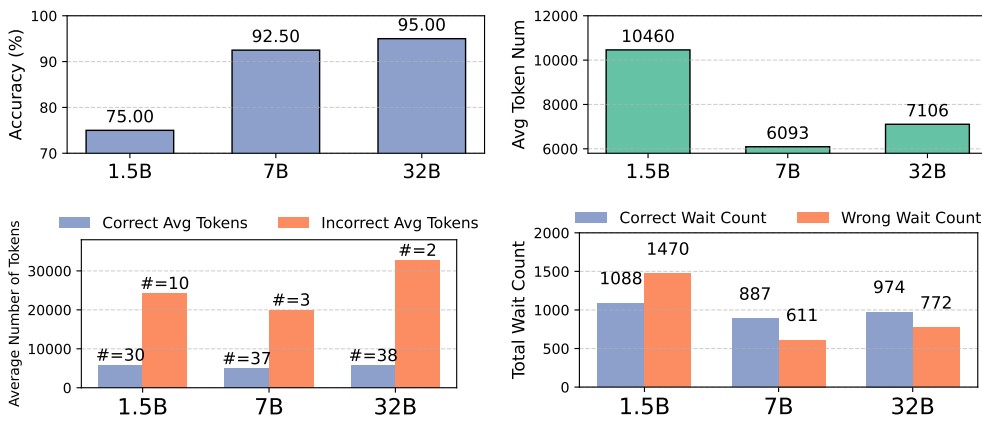

Figure 11: Accuracy and output statistics of three models on the AMC23 dataset.

- **Edge-Cloud Hybrid Systems.** In latency- or bandwidth-sensitive applications (e.g., mobile or offline settings), a 1.5B model can operate locally, invoking a remote 32B model only when necessary. This design reduces cloud compute usage while maintaining reasoning capability.

These examples demonstrate how speculative thinking can be flexibly integrated into real-world systems that must balance performance, cost, and scalability.

