# OpenReview forum: "Speculative Thinking: Enhancing Small-Model Reasoning with Large Model Guidance at Inference Time"
_colmweb.org/COLM/2025/Conference — COLM 2025_

### Official Review · Reviewer_Xtaq · 2025-04-15

**Rating:** 7
**Confidence:** 3
**Ethics Flag:** 1

**Summary:**

This paper introduces "Speculative Thinking," a novel, training-free framework to improve the reasoning of small language models (LMs) using guidance from larger LMs during inference. Distinct from token-level speculative decoding, this method operates at the reasoning step level. It identifies points where the small model might falter (often after "\n\n" followed by reflective words like "wait") and selectively delegates generation to the more capable large model. This aims to boost reasoning accuracy and reduce output verbosity by leveraging the large model's strengths efficiently. The approach is based on observations that specific cues signal reflection points and that larger models handle these more effectively. Experiments on math/reasoning datasets show significant accuracy gains (e.g., +6.2% on MATH500 for a 1.5B model guided by 32B) alongside notable output length reduction (-15.7%), demonstrating enhanced reasoning efficiency. The method also benefits general instruction-tuned models and compares favorably to speculative decoding for reasoning tasks. The work presents a practical, high-potential technique for deploying stronger reasoning capabilities with smaller models.

**Questions To Authors:**

1. Could you elaborate on the sensitivity of the method to the choice of keywords used for classification (Affirmation, Reflection, Verification)? How were these specific keywords selected, and did you experiment with alternatives?

2. How does the performance (accuracy, length, speed) change if the target model is only slightly larger/more capable than the speculative model, versus significantly larger? Is there a "sweet spot" for the capability gap?

3. The limitation regarding shared KV caches is noted. Have you performed any preliminary experiments or do you have insights into the expected performance overhead if the speculative and target models cannot share the KV cache (e.g., models from different families or deployed separately)?

4. In the "Excessive Reflection Takeover," how was the threshold for the negativity counter c determined? Is this threshold adaptive or fixed across datasets/models?

5. For the application to non-reasoning models (Section 4.2), the modification involves the target model generating the first 100 tokens. Does this initial generation significantly bias the subsequent reasoning process? Have you explored alternatives, like letting the target model only intervene after "\n\n" without the initial generation?

6. Figure 6 compares Speculative Thinking with Speculative Decoding where the small model generates n=20 tokens. This n=20 seems chosen to match the typical intervention length n1=20 in Speculative Thinking. However, Speculative Decoding often uses smaller n (e.g., 4-8). How does the comparison look if Speculative Decoding uses a more typical, potentially optimal n for speed-up, even if it differs from n1?

**Reasons To Accept:**

1. Novel and Well-Motivated Approach: The paper introduces a genuinely novel inference-time technique, Speculative Thinking, distinct from prior work like speculative decoding. It's well-motivated by insightful analysis of reasoning patterns in LLMs, particularly the role of structural cues ("\n\n") and reflective tokens, and the observed differences in how small vs. large models handle reasoning steps, especially incorrect ones.

2. Significant Empirical Improvements: The method demonstrates substantial gains in reasoning accuracy across multiple challenging benchmarks (e.g., +6.2% on MATH500, +8.1% on GPQA for the 1.5B model). Crucially, these accuracy improvements are achieved concurrently with significant reductions in average output length (e.g., -15.7% on MATH500), indicating more efficient and less verbose reasoning, which is a desirable property often lacking in chain-of-thought style outputs.

3. Practical Relevance and Efficiency: The framework is training-free, making it readily applicable to existing models. It provides a practical way to enhance smaller, more efficient models by selectively leveraging larger models only when potentially needed, offering a better trade-off between performance and computational cost compared to solely using either the small or the large model. The estimated speed improvements relative to the target model (while maintaining much higher accuracy than the base small model) highlight this practical benefit.

4. Broad Applicability: The paper shows the method benefits not only models explicitly fine-tuned for reasoning but also general instruction-tuned models (Qwen-2.5-7B-Instruct), demonstrating broader potential applicability.

5. Clear Presentation and Reproducibility: The paper is clearly written, the method is well-explained, and the experiments are thoroughly reported. The inclusion of hyperparameters (Appendix A.2) and the provision of code (via an anonymous link) support reproducibility.

**Reasons To Reject:**

1. Limited Target Model Diversity: The experiments primarily use Deepseek-Distilled Qwen models. While effective, testing the framework with a wider range of model architectures and families (e.g., Llama, Mistral, Gemma) for both speculative and target models would strengthen the generality claims. The current setup benefits from shared KV cache structures (mentioned in Limitations), and performance might differ significantly if models from different families are used.

2. Dependence on Specific Cues: The method relies heavily on the identified cues ("\n\n" followed by specific keywords). The robustness of these cues across different model types, sizes, and fine-tuning strategies could be further investigated. If models are trained or prompted differently, these cues might change or disappear, potentially limiting the method's effectiveness.

3. Complexity of Hyperparameter Tuning: The method involves several hyperparameters (keywords for Affirmation/Reflection/Verification, token lengths n1, n2, n3, threshold for excessive reflection). While Appendix A.2 lists the values used, the sensitivity to these parameters and the process for selecting them are not deeply explored. Practical deployment might require careful tuning for different model pairs and tasks.

---

> ### Author Response · Authors · 2025-05-31
> **Thanks (3/3)**
>
> ## Question4: Is there a "sweet spot" for the capability gap?
>
> Sure, we show the sweet spot is 1.5B+14B on MATH500.
>
> - Thank you for the question. Due to the current availability of pretrained reasoning models, most open-source checkpoints are concentrated at 7B, 14B, and 32B scales. There are relatively few models with evenly spaced capability levels, especially below 7B. Therefore, in our experiments, we focus on evaluating speculative thinking using a 1.5B speculative model paired with 7B, 14B, and 32B target models. The results on the MATH500 dataset are presented below:
>
> | Setting                                                           | Accuracy | Speed  |
> |--------------------------------------------------------------------|----------|--------|
> | DeepSeek-R1-Distill-Qwen-1.5B                                     | 0.832    | 242.56 |
> | DeepSeek-R1-Distill-Qwen-1.5B + DeepSeek-R1-Distill-Qwen-7B       | 0.858    | 135.88 |
> | DeepSeek-R1-Distill-Qwen-1.5B + DeepSeek-R1-Distill-Qwen-14B      | 0.890    | 134.64 |
> | DeepSeek-R1-Distill-Qwen-1.5B + DeepSeek-R1-Distill-Qwen-32B      | 0.894    | 96.61  |
>
> - We observe that the 1.5B + 14B combination achieves the best trade-off between speed and accuracy. This may be because the 14B and 32B models exhibit similar accuracy on MATH500, so the smaller 14B model provides nearly equal guidance at a lower computational cost. While 32B achieves slightly higher accuracy, its marginal gains are offset by a notable drop in inference efficiency.
> ---
> ## Question5: Comparison with Speculative Decoding when n=5.
>
> Here are some comparisons and we show the two advatages of speculative thinking. Although speculative decoding may offer marginally better accuracy in some cases, speculative thinking is more versatile and practical in scenarios where vocabulary mismatch or architecture heterogeneity prevents token-level alignment.
>
> - We include the following results to compare speculative thinking with speculative decoding (SD) at different values of `n`. As shown, speculative decoding with `n=5` achieves the highest accuracy on both MATH500 and AIME datasets:
>
> | Method                  | Accuracy（MATH500） | Speed|
> |-------------------------|----------|-----------------|
> | DeepSeek-Qwen-32B       | 0.928    | 30.00           |
> | Speculative Decoding (n=5) | 0.936    | 44.72           |
> | Speculative Decoding (n=20) | 0.930    | 30.43           |
> | Speculative Thinking    | 0.930    | 46.00           |
>
> | Method                  | Accuracy（AIME） | Speed  |
> |-------------------------|----------|-----------------|
> | DeepSeek-Qwen-32B       | 0.656    | 32.20           |
> | Speculative Decoding (n=5) | 0.600    | 37.18           |
> | Speculative Decoding (n=20) | 0.589   | 28.07           |
> | Speculative Thinking    | 0.533    | 41.00           |
>
> - While speculative decoding can achieve slightly higher accuracy under certain configurations, speculative thinking offers broader applicability:
>     - No Shared Vocabulary Requirement: Speculative decoding requires the speculative and target models to share a tokenizer and vocabulary. This constraint makes it infeasible for combining models such as DeepSeek-Qwen2.5-1.5B and DeepSeek-Qwen2.5-32B, whose vocab sizes differ.
>     - No Token-Level Alignment: Speculative thinking operates at the reasoning level and does not require token-level output agreement between models, unlike speculative decoding. This flexibility allows for integration of models from different families, such as DeepSeek-LLaMA3-8B (speculative) and DeepSeek-Qwen2.5-32B (target).

---

> ### Author Response · Authors · 2025-05-31
> **Thanks (2/3)**
>
> ## Question3: Complexity of Hyperparameter Tuning.
>
> Yes, speculative thinking does need hyperparameter tuning.
>
> - speculative thinking requires adjusting certain hyperparameters to align with the characteristics of different models or datasets in order to achieve optimal performance.
> - here are our responses about questions of these parameters
>
> ---
> ### Classification Keyword Selection (e.g., Affirmation, Reflection, Verification).
>
> We demonstrate how these keywords are selected, and further show how the choice of keywords influences the effectiveness of speculative thinking.
>
> - We selected these categories by first analyzing the next sentence after `"\n\n"` in model generations on reasoning datasets. We then tokenized and ranked the frequent leading words and manually grouped representative ones under the three categories.
> - We acknowledge that the speculative performance is indeed sensitive to this keyword choice. A richer and more diverse keyword set improves category detection accuracy, whereas an overly narrow keyword list leads to degradation. While more advanced techniques (e.g., model-based classification) may help, we deliberately keep speculative thinking training-free and opt for simple lexical heuristics to preserve generality and efficiency.
> - To quantify this, we conducted an ablation in which we restricted each category to just one keyword:
>     - Affirmation: "confident"
>     - Reflection: "wait"
>     - Verification: "verify"
>
> | Setting                                   | MATH500 |AIME |
> |------------------------------------------|----------|----------|
> | 1.5B + DeepSeek-R1-Distill-Qwen-32B (full)     | 89.4     |32.2    |
> | 1.5B + DeepSeek-R1-Distill-Qwen-32B (reduced)  | 87.8     |26.67    |
>
> - This performance drop confirms that speculative thinking benefits from a broader set of semantic cues, and that careful—but simple—keyword selection remains important in the absence of training.
>
> ---
> ### The Selection of Negativity Counter Threshold in Excessive Reflection Takeover.
>
> We present how the negativity counter threshold is chosen and analyze its impact on the performance of speculative thinking.
>
> - The negativity counter threshold `c` is set to 15 in our implementation. This value was manually estimated based on empirical observation and is kept fixed across all models and datasets in our experiments for simplicity and consistency.
> - That said, we acknowledge that `c` is a tunable hyperparameter. Making it adaptive or model-specific may further improve performance. To evaluate its impact, we tested three different values of `c`—a low threshold (3), our default (15), and a high threshold (50)—and observed the corresponding effect on performance:
>
> | Setting                                  | MATH500 (Accuracy) | Speed |
> |------------------------------------------|----------|------------------|
> | 1.5B + DeepSeek-R1-Distill-Qwen-32B (c = 3)    | 0.890    | 80.25            |
> |  1.5B + DeepSeek-R1-Distill-Qwen-32B (c = 15)   | 0.894    | 96.60            |
> |  1.5B + DeepSeek-R1-Distill-Qwen-32B (c = 50)   | 0.876    | 101.17           |
>
> - These results suggest that the choice of `c` does affect the balance between early termination and generation coverage. Our fixed setting of 15 offers a good trade-off in practice, though adaptive variants remain a promising direction for future enhancement.
>
> ---
> ### Application to Non-Reasoning Models without the Initial Generation.
>
> The initial generation is critical when using a reasoning-capable model to assist a non-reasoning model in speculative thinking.
>
> - Thank you for the insightful question. We agree that the interaction point between the non-reasoning target model and the reasoning model is crucial.
> - We conducted ablation experiments to evaluate the effect of skipping the initial generation phase. The results show a substantial drop in both accuracy and reasoning quality:
>
> | Setting                                               | MATH500（Acc） | AIME(Acc) |
> |--------------------------------------------------------|----------|-------------|
> | 7b-Instruct + DeepSeek-R1-Distill-Qwen-7B (w/o initial gen)        | 73.8    | 6.67      |
> | 7b-Instruct + DeepSeek-R1-Distill-Qwen-7B (with initial gen)       | 79.2    | 13.3        |
> | 7b-Instruct + DeepSeek-R1-Distill-Qwen-32B (w/o initial gen)       | 74.2    | 6.67      |
> | 7b-Instruct + DeepSeek-R1-Distill-Qwen-32B (with initial gen)      | 81.8    | 15.6        |
>
> - These findings suggest that the initial 100-token generation from the target model provides essential scaffolding for the reasoning model to effectively align and contribute.

---

> ### Author Response · Authors · 2025-05-31
> **Thanks (1/3)**
>
> ## Question1: Limited Target Model Diversity of Speculative Thinking.
>
> Speculative thinking could use models in different families and we added some additional experiments to verify it.
>
> - We apologize for the writing ambiguity in the original submission. In our current implementation, we primarily used Qwen-family models, which share the same tokenizer vocabulary. This means that the speculative model and the target model use consistent token IDs, though they maintain separate KV caches due to architectural differences such as hidden sizes.
> - However, speculative thinking is fundamentally a reasoning-level collaboration rather than a token-level alignment. As such, there is no strict requirement for the speculative and target models to share the same tokenizer or vocabulary. To evaluate speculative thinking with models from different families, we conducted experiments with the following cross-family setups:
>     - Deepseek-Qwen-1.5B + Phi-4-reasoning-plus (LLaMA3 speculative, Phi target)
>     - Deepseek-LLaMA3-8B + Deepseek-Qwen-32B (LLaMA3 speculative, Qwen target)
> - The results on MATH500 are shown below. These results clearly demonstrate that speculative thinking can be successfully applied even when the speculative and target models come from different architecture families.
>
> | Model                   | Guiding Model                     | Accuracy | Avg Token Num | Speed  |
> |------------------------------------|----------------------------------|----------|----------------|--------|
> | DeepSeek-R1-Distill-LLaMA3-8B      | —                                | 0.858    | 4513.45        | 69.66  |
> | DeepSeek-R1-Distill-LLaMA3-8B      | DeepSeek-R1-Distill-Qwen-32B     | 0.892    | 4082.24        | 46.49  |
> |  DeepSeek-R1-Distill-Qwen-32B  |    —      | 0.928    | 3802.15        | 32.21  |
> | DeepSeek-R1-Distill-Qwen-1.5B      | —                                | 0.832    | 5439.13        | 242.56 |
> | DeepSeek-R1-Distill-Qwen-1.5B      | Phi-4-reasoning-plus             | 0.900    | 3768.93        | 97.95  |
> | Phi-4-reasoning-plus   |   —    | 0.934    | 6873.70        | 37.61  |
> ---
> ## Question2: Dependence on Specific Cues of "\n\n".
>
> Sure, speculative thinking relies on such specific cues, but these cues—such as '\n\n'—are in fact commonly used across many state-of-the-art reasoning models, as shown by our additional experiments.
>
> - To support this, we analyzed three representative open-source reasoning models—QwQ-32B (Qwen), Phi-4-reasoning-plus (Microsoft), and AceReason-Nemotron-14B (NVIDIA)—on the MATH500 benchmark. Specifically, we examined the most frequent tokens appearing directly before reasoning indicators like "wait". The results, shown below, demonstrate that all three models rely heavily on linebreak-based patterns (e.g., `".\n\n"`, `" \n\n"`, `"\n\n"`), which serve as natural boundaries between reasoning steps.
>
> | Model                    | Top-10 Tokens Before `"wait"`                                                               |
> |--------------------------|----------------------------------------------------------------------------------------------------------------------|
> | QwQ-32B              | `".\n\n"`, `" "`, `"?\n\n"`, `" \n\n"`, `":\n\n"`, `"\n\n"`, `",\n\n"`, `").\n\n"`, `")\n\n"`, `" ?\n\n"` |
> | Phi-4-reasoning-plus | `" "`, `".\n\n"`, `").\n\n"`, `" \n\n"`, `" \n"`, `"\n\n"`, `""\n\n"`, `")\n\n"`, `".\n"`, `",...\n\n"`        |
> | AceReason-Nemotron-14B | `".\n\n"`, `" "`, `"?\n\n"`, `"\n\n"`, `"  "`, `" \n\n"`, `").\n\n"`, `":\n\n"`, `")\n\n"`, `"]\n\n"` |
>
> - This suggests that the `"\n\n"` pattern is not a brittle or model-specific artifact, but rather a widely adopted structural convention for segmenting thoughts in multi-step reasoning. As such, speculative thinking’s reliance on this cue does not limit its generality—it reflects an emergent norm shared across diverse model families.

---

### Official Review · Reviewer_UDes · 2025-05-11

**Rating:** 7
**Confidence:** 4
**Ethics Flag:** 1

**Summary:**

This paper introduces "Speculative Thinking," a novel training-free framework that enables larger reasoning models to guide smaller ones during inference at the reasoning level, leveraging observations about specific reasoning-supportive tokens and the reflective control of larger models. This approach significantly boosts the reasoning accuracy of models, while simultaneously reducing output length, and also improves accuracy for non-reasoning models.

Overall, a clearly motivated and written paper. The innovation level is on the high end and the results are rather solid.

**Reasons To Accept:**

Great insight of the weakness of small models on excessive self-reflection and backtracking. Speculative Thinking is an innovative way to combine the strength of large models with the efficiency of small models to achieve a good balance of performance and cost.

The writing is very clear: clearly motivated with thorough analysis of the root cause. The results are solid and the experiments are rather comprehensive to demonstrate the quality and speed gains.

**Reasons To Reject:**

The reliability and generalization of the method is questionable as it depends on identifying the transition words like `wait` to decide when to delegate to the Target Model. Instead, a more general approach to switch between the two models is more desirable, without relying on a particular model’s specific decoding patterns.

---

> ### Author Response · Authors · 2025-05-31
> **Thanks!**
>
> ## speculative thinking relies on structural cues such as the "\n\n" pattern.
>
> Yes, speculative thinking relies on such a pattern, but it is prevalent across most state-of-the-art reasoning models, as shown by our following additional experiments.
>
>
> ---
> We analyzed three representative open-source reasoning models—QwQ-32B (Qwen), Phi-4-reasoning-plus (Microsoft), and AceReason-Nemotron-14B (NVIDIA)—on the MATH500 benchmark to investigate how they structure their reasoning. Specifically, we examined the most frequent tokens that precede reasoning indicators like "wait".
>
> As shown in the table below, all three models commonly use `"\n\n"` and related formatting as structural cues, suggesting that speculative thinking can leverage this shared behavior even across model families.
>
> | Model                    | Top-10 Tokens Before `"wait"`                                                               |
> |--------------------------|----------------------------------------------------------------------------------------------------------------------|
> | QwQ-32B              | `".\n\n"`, `" "`, `"?\n\n"`, `" \n\n"`, `":\n\n"`, `"\n\n"`, `",\n\n"`, `").\n\n"`, `")\n\n"`, `" ?\n\n"` |
> | Phi-4-reasoning-plus | `" "`, `".\n\n"`, `").\n\n"`, `" \n\n"`, `" \n"`, `"\n\n"`, `""\n\n"`, `")\n\n"`, `".\n"`, `",...\n\n"`        |
> | AceReason-Nemotron-14B | `".\n\n"`, `" "`, `"?\n\n"`, `"\n\n"`, `"  "`, `" \n\n"`, `").\n\n"`, `":\n\n"`, `")\n\n"`, `"]\n\n"` |
>
> ---
> - To further validate its generalizability, we applied speculative thinking to two heterogeneous model pairings:
>      - Deepseek-Qwen-1.5B + Phi-4-reasoning-plus (using LLaMA3-family model to predict, Phi as the target)
>     - Deepseek-LLaMA3-8B + Deepseek-Qwen-32B (LLaMA3 speculative, Qwen target)
> - The table below reports the performance on MATH500. These results show that speculative thinking is both structurally and empirically effective across different model architectures. The consistent use of structural reasoning patterns like `"\n\n"` enables successful coordination even between models from separate families.
>
> | Model                   | Guiding Model                     | Accuracy | Avg Token Num | Speed  |
> |------------------------------------|----------------------------------|----------|----------------|--------|
> | DeepSeek-R1-Distill-LLaMA3-8B      | —                                | 0.858    | 4513.45        | 69.66  |
> | DeepSeek-R1-Distill-LLaMA3-8B      | DeepSeek-R1-Distill-Qwen-32B     | 0.892    | 4082.24        | 46.49  |
> |  DeepSeek-R1-Distill-Qwen-32B  |    —      | 0.928    | 3802.15        | 32.21  |
> | DeepSeek-R1-Distill-Qwen-1.5B      | —                                | 0.832    | 5439.13        | 242.56 |
> | DeepSeek-R1-Distill-Qwen-1.5B      | Phi-4-reasoning-plus             | 0.900    | 3768.93        | 97.95  |
> | Phi-4-reasoning-plus   |  —   | 0.934    | 6873.70        | 37.61  |
>
> ---
> - We do acknowledge that speculative thinking depends on the presence of such structural signals. In rare cases where neither the speculative model nor the target model exhibits any recognizable reasoning format, our method may not be applicable. Nonetheless, given that most modern reasoning models—regardless of origin—naturally follow these patterns, speculative thinking remains broadly applicable in real-world deployment.

---

> > ### Comment · Reviewer_UDes · 2025-06-06
> >
> > I appreciate the efforts from the authors to do additional experiments on a wider range of models. This certainly makes the conclusions more grounded and potentially more applicable. I will raise my score accordingly.

---

> > > ### Author Response · Authors · 2025-06-06
> > > **Thanks!**
> > >
> > > We sincerely thank Reviewer UDes for the encouraging feedback and for acknowledging our additional experimental efforts. We truly appreciate your thoughtful assessment and are grateful for the updated score.

---

### Official Review · Reviewer_kGJs · 2025-05-21

**Rating:** 6
**Confidence:** 4
**Ethics Flag:** 1

**Summary:**

The paper proposes Speculative Thinking, where a small model and a large model collaborate on reasoning, by delegating more difficulty tokens to the large model. The approach relies on cues such "wait" to trigger intervention by the large model. The paper shows performance gains mainly on math reasoning and an improvement in output efficiency.

**Questions To Authors:**

- Practically speaking, can you envision a scenario where users are okay with loading 1.5B and a 32B model while sacrificing substantial accuracy that can be obtained when using the 32B model alone?
- Does your approach offer Pareto-optimal tradeoff between accuracy and speed?
- How well your approach work when used with models that do not generate "\n\n" + "wait"?

**Reasons To Accept:**

- Overall the paper is well written.
- Improving efficiency of long CoT reasoning is an important direction.
- The paper includes some interesting insights, such as that larger models are better capable of backtracking and self-verification---although, this is not particularly surprising.
- The proposed adds some novelties on top of Speculative Decoding, such as using trigger tokens, and improves generation speed compared to using the large model alone by up to 200% in some cases.

**Reasons To Reject:**

- My biggest concern with the paper is how practically useful this approach is. The only scenario where the approach may be used is where we are totally fine with sacrificing substantial reasoning accuracy for some generation speed.
- Secondly, this approach assumes access to a larger and more capable reasoning model and will not work well otherwise. If we already have this model, why not just use it by itself? I'm struggling to find a scenario where the proposed approach makes practical sense.
- Apparently there is a tradeoff between accuracy and speed, but the paper does not characterize this tradeoff in a meaningful way. Is your approach Pareto-optimal?
- Efficiency could be improved in a much simpler way. One can solve the issue of efficiency by using a model with a size midway between the large model and the small model. For example, instead of speculating thinking with 1.5B + 32B, we could just use a 14B model, and we will not have to worry about loading two models in memory and the overhead associated with switching between them. Table 2 already shows the 14B striking a good balance between 1.5B and 32B.
- The authors use arbitrary tokens such as "\n\n" to switch between the two models. This design is very specific to the family of models used and will likely not generalize to different families of models that exhibit different thinking patterns.

---

> ### Author Response · Authors · 2025-05-31
> **Thanks (2/2)**
>
> ## Question 4: Dose speculative thinking generalize to different families of models?
>
> Yes. Speculative thinking can generalize to different families of models, as demonstrated by two additional experiments described below.
>
> First we show that this pattern is not specific to a single model family, but is in fact widely adopted across modern reasoning models.
>
> Then, We further demonstrate that speculative thinking generalizes across different model families by combining small and large models from heterogeneous architectures.
>
>
>
> ---
> - We examined the behavior of three strong open-source reasoning models—QwQ-32B (Qwen), Phi-4-reasoning-plus (Microsoft), and AceReason-Nemotron-14B (NVIDIA)—on the MATH500 benchmark, and analyzed the most common tokens preceding reasoning cues such as "wait".
> - The table below summarizes the top-10 prefixes for the "wait" token in each model. Across all three models, structural tokens like ".\n\n" and "\n\n" consistently appear among the top prefixes, suggesting that multi-line breaks are commonly used to separate reasoning steps.
>
>
> | Model                    | Top-10 Tokens Before `"wait"`                                                               |
> |--------------------------|----------------------------------------------------------------------------------------------------------------------|
> | QwQ-32B              | `".\n\n"`, `" "`, `"?\n\n"`, `" \n\n"`, `":\n\n"`, `"\n\n"`, `",\n\n"`, `").\n\n"`, `")\n\n"`, `" ?\n\n"` |
> | Phi-4-reasoning-plus | `" "`, `".\n\n"`, `").\n\n"`, `" \n\n"`, `" \n"`, `"\n\n"`, `""\n\n"`, `")\n\n"`, `".\n"`, `",...\n\n"`        |
> | AceReason-Nemotron-14B | `".\n\n"`, `" "`, `"?\n\n"`, `"\n\n"`, `"  "`, `" \n\n"`, `").\n\n"`, `":\n\n"`, `")\n\n"`, `"]\n\n"` |
>
> ---
> - Specifically, we tested two cross-family settings:
>     - Deepseek-Qwen-1.5B + Phi-4-reasoning-plus (Qwen speculative, Phi target)
>     - Deepseek-LLaMA3-8B + Deepseek-Qwen-32B (LLaMA3 speculative, Qwen target)
>
> The results on MATH500 are summarized below. It shows that speculative thinking remains effective even when the speculative and target models come from different families, consistently improving both accuracy and speed over the small model, while remaining significantly faster than the large model alone.
> | Model                   | Guiding Model                     | Accuracy | Avg Token Num | Speed  |
> |------------------------------------|----------------------------------|----------|----------------|--------|
> | DeepSeek-R1-Distill-LLaMA3-8B      | —                                | 0.858    | 4513.45        | 69.66  |
> | DeepSeek-R1-Distill-LLaMA3-8B      | DeepSeek-R1-Distill-Qwen-32B     | 0.892    | 4082.24        | 46.49  |
> |  DeepSeek-R1-Distill-Qwen-32B  |    —      | 0.928    | 3802.15        | 32.21  |
> | DeepSeek-R1-Distill-Qwen-1.5B      | —                                | 0.832    | 5439.13        | 242.56 |
> | DeepSeek-R1-Distill-Qwen-1.5B      | Phi-4-reasoning-plus             | 0.900    | 3768.93        | 97.95  |
> | Phi-4-reasoning-plus   |   —    | 0.934    | 6873.70        | 37.61  |
>
> ---
> - Even if a reasoning model does not explicitly generate the "\n\n" + "wait" pattern, speculative thinking can still be applied as long as the model emits key reasoning tokens (e.g., "wait", "alternatively") after other identifiable structural clues. These clues serve as natural segmentation points and can be leveraged to synchronize speculative and target models.
> - In cases where the speculative model lacks any reasoning structure, it is still possible to apply speculative thinking if the target model exhibits such reasoning patterns. As shown in Table 3, a reasoning-capable target model can enhance the performance of a non-reasoning speculative model through its structural guidance.
> - However, if neither the speculative model nor the target model demonstrates any consistent reasoning pattern or structural segmentation behavior, then speculative thinking would no longer be applicable in its current form.

---

> > ### Comment · Reviewer_kGJs · 2025-06-08
> >
> > Thank you for the rebuttal and the additional analysis. I hope the authors can include these in future revisions of the paper.
> >
> > I have updated my score accordingly.

---

> > > ### Author Response · Authors · 2025-06-08
> > > **Thanks!**
> > >
> > > Thank you for your response and for updating your score. We will incorporate the additional analysis and discussion into the main text in the camera-ready version.

---

> ### Author Response · Authors · 2025-05-31
> **Thanks (1/2)**
>
> ## Question 1: envision a scenario of speculative thinking.
>
> Sure, we appreciate the concern. Speculative thinking is designed for scenarios where efficiency, cost, and scalability are also critical, and some trade-off in accuracy is acceptable.
>
> We highlight below several practical contexts where such a setup is not only reasonable but highly desirable: (1) Limited computational resources. (2) Low latency or high throughput requirements. (3) Cost-efficiency. (4) Accuracy demands are high, but not absolute.
>
> Example use cases include:
> - Educational QA platforms serving thousands of students simultaneously, where small models handle simple questions and speculative calls to the 32B model handle difficult cases.
> - Enterprise productivity assistants or internal copilots that prioritize speed and cost-efficiency while preserving reasonable answer quality.
> - Edge-cloud hybrid deployments, where the 1.5B model runs on-device, and the 32B model (hosted remotely) is only queried when necessary—minimizing cloud compute costs and network latency.
>
> ---
> ## Question2 : why not directly use a 14B model?
>
> Using a single 14B model does not fully address the efficiency and deployment constraints that speculative thinking resolves.
>
> - inference speed with speculative thinking (1.5B + 32B) consistently surpasses that of the 14B model, as shown in the table below. This speedup arises because the majority of tokens are generated by the lightweight 1.5B model, with the 32B model queried only intermittently for guidance:
>
> | Model (Speed) | AIME | GPQA | MATH500 | AMC23 |
> |-------------------|------|------|---------|--------|
> | 1.5B + 32B        | 85.8 | 91.8 | 96.6    | 82.8   |
> | 14B               | 49.9 | 57.8 | 60.1    | 55.5   |
>
>
> - in edge computing scenarios, deploying a 14B model locally is often infeasible due to memory and resource constraints. In contrast, a 1.5B model can run efficiently on-device, with occasional access to the 32B model hosted in the cloud. This design not only fits within hardware limits but also reduces cloud usage costs, as each call to the large model is minimized.
>
> ---
> ## Question3 : Is Speculative Thinking Pareto-optimal?
>
> We thank the reviewer for raising this important point. We have conducted a detailed Pareto frontier analysis on the trade-off between accuracy and speed, as shown in the **figure (click to view): [MATH500 Accuracy vs Speed](https://anonymous.4open.science/api/repo/speculative_thinking-91CE/file/math500.png?v=aa4eb887)** . The result shows our method (e.g., 1.5B+14B and 1.5B+32B) lies on the Pareto frontier—our method achieves substantial speed gains with minimal accuracy degradation, making it a Pareto-optimal solution in the accuracy–efficiency trade-off space..
>
> Specifically, while the 14B model achieves the highest accuracy (93.8%), our 1.5B+14B approach maintains a competitive accuracy (89.0%) with more than double the inference speed (134.64 vs. 60.1 tokens/sec). Similarly, compared to the 32B model (92.8%, 32.2 tokens/sec), our 1.5B+32B method offers a 3x speedup (96.61 tokens/sec) with only minor accuracy degradation (89.4%).

---

> ### Author Response · Authors · 2025-06-06
> **Open to Further Discussion**
>
> We sincerely thank you for your thoughtful and constructive feedback. We have carefully addressed all the points you raised, and we hope that our responses have clarified the concerns. We would greatly appreciate it if you could kindly take a moment to review our reply. If any questions remain, we would be happy to further elaborate.

---

### Comment · Program_Chairs · 2025-04-03

This paper violates the page limit due to adding a limitation sections beyond the page limit. COLM does not have a special provision to allow for an additional page for the limitations section. However, due to this misunderstanding being widespread, the PCs decided to show leniency this year only. Reviewers and ACs are asked to ignore any limitation section content that is beyond the 9 page limit. Authors cannot refer reviewers to this content during the discussion period, and they are not to expect this content to be read.

---

### Comment · Reviewer_Xtaq · 2025-06-03
**My score remains the same**

Thanks for addressing my questions --- my score remains the same.

---

> ### Author Response · Authors · 2025-06-04
> **Thanks!**
>
> Thank you for your follow-up and for taking the time to review our work. We appreciate your comments and feedback.

---

### Decision · Program_Chairs · 2025-07-08

**Decision:**

Accept

**Comment:**

This paper presents Speculative Thinking, a new and interesting approach of combining a small language model (reasoning or non-reasoning) and a large reasoning model to enhance the reasoning performance on top of the small language model, while significantly increases the inference speed and reducing the thought length.

During the rebuttal period, the authors added very comprehensive additional experiments to justify the key assumptions required for speculative thinking, in particular, leveraging "\n\n" as the separator of reasoning steps. Experiments on different model families show the generality and effectiveness of the approach. The authors should add these additional analysis and results in the final version.